

# Genomic assembly, characterization, and quantification of DICER-like gene family in Okra plants under dehydration conditions

Hagar Tarek Elhefnawi[1,*], Mohamed Abdel Salam Rashed[1], Ayman Atta[1], Rana M. Alshegaihi[2], Khairiah Mubarak Alwutayd[3], Diaa Abd El-Moneim[4] and Mahmoud Magdy[1,*]

[1] Department of Genetics, Faculty of Agriculture, Ain Shams University, Cairo, Egypt
[2] Department of Biology, College of Science, University of Jeddah, Jeddah, Saudi Arabia
[3] Department of Biology, College of Science, Princess Nourah bint Abdulrahman University, Riyadh, Saudi Arabia
[4] Department of Plant Production (Genetic Branch), Faculty of Environmental Agricultural Sciences, Arish University, El-Arish, Egypt
[*] These authors contributed equally to this work.

Corresponding author
Mahmoud Magdy,
m.elmosallamy@agr.asu.edu.eg

## ABSTRACT

**Background.** Okra is a plant farmed for its pods, leaves, and stems all of which are edible. It is famous for its ability to tolerate long desiccation periods. It belongs to the Malvaceae family and is a sister species to hibiscus, cotton, and cacao plants.

**Methods.** In the current study, okra plants were used as a model to sequence, assemble, and analyze the evolutionary and functional characteristics of the Dicer-like protein gene family (DCL) based on DNAseq and qPCR techniques.

**Results.** Four Dicer-like (DCL) single-copy genes of the okra plant *Abelmoschus esculentus* (L.) Moench (AeDCL) were successfully assembled. The lengths of the AeDCL copies were 8,494, 5,214, 4,731, and 9,329 bp. The detected exons in these samples ranged from a single exon in *AeDCL3* to 24 exons in *AeDCL4*. AeDCLs had five functional domains of two DEAD-like helicase superfamilies, N and C; one Dicer domain; one ribonuclease III domain (a and b); and one double-stranded RNA-binding domain. The PAZ domain was completely annotated only for *AeDCL1* and *AeDCL3*. All AeDCLs were up-regulated under drought conditions, with leaves showing more extensive fold changes than roots. The study focused on a comprehensive genome-wide identification and analysis of the DCL gene family in naturally drought-tolerant okra plants, an orphan crop that can be used as a model for further genomic and transcriptomic studies on drought-tolerance mechanisms in plants.

## INTRODUCTION

The cultivated okra plant *Abelmoschus esculentus* (L.) Moench belongs to the family Malvaceae and is considered one of the important pod vegetable crops in developing countries, mainly in Asia and Africa (*Gemede et al., 2015*; *Gemede et al., 2016*). The total

amount of okra produced in the world is approximately 10.5 million tons of pods with high mucilage. India contributes 70% of this total, followed by Nigeria with approximately 17.4% and Egypt with 1.7% (*Food and Agriculture Organization, 2022*). According to reports, the okra plant is one of the orphan crops of Africa (*i.e.,* it is a nutritional and medicinal indigenous crop that is often cultivated and widely accepted by local farmers). It is known for its relative drought tolerance (*Naveed, Khan & Rauf, 2012*; *Singh et al., 2014*); however, severe droughts can reduce its pod yields by up to 70%, especially if they occur during the stage of flowering or pod formation (*Chaturvedi et al., 2019*; *Romaisa et al., 2015*). Drought is one of the most severe abiotic stresses in the world, and it can reduce the growth, development, and yield of plants (*Gong et al., 2005*; *Kusvuran, 2012*). Fortunately, plants sometimes develop distinct mechanisms that allow them to cope with drought stress; these mechanisms may be associated with their morphologic, physiologic, and biochemical functions (*Farooq et al., 2009a*; *Farooq et al., 2009b*). Depending on the intensity and duration of the water stress, as well as the plant species and its growth stage, the responses of plants to water stress differ significantly at various organizational levels. A number of signal transduction pathways are involved in the acquisition of stress tolerance in plants (*Shinozaki, Yamaguchi-Shinozaki, & Seki, 2003*). In the eukaryotic cell, a mechanism regulates gene expression by generating small RNAs and utilizing those RNAs either during or after transcription (*Finnegan & Matzke, 2003*). In eukaryotes, small RNAs (19 to 28 nt in length) involved in RNA silencing have been classified as (1) micro-RNAs (miRNAs) generated from the double-stranded RNA (dsRNA) region of hairpin-shaped precursor RNAs or (2) small interfering RNAs (siRNAs) derived from long dsRNAs (*Finnegan & Matzke, 2003*; *Bartel, 2004*). Dicer or Dicer-like (DCL) proteins are large RNaseIII-like enzymes responsible for cleaving these templates into small RNAs (*Bernstein et al., 2001*). There is growing evidence that DCL proteins are important players in RNA silencing, the regulation of gene expression through miRNAs, and the protection of plants from viral infections (*Moura et al., 2019*; *Belal et al., 2022*). They form a small protein family in plants whose diversification dates to the emergence of the moss *Physcomitrella patens*. Despite the DCL widespread expression, it may differ depending on the tissue type and developmental stage of a plant and on the environmental stress to which the plant is exposed (*Liu, Feng & Zhu, 2009*; *Belal et al., 2022*).

The number of DCL gene copies varies among plants. For instance, 11 DCL genes have been identified in the cotton plant *Gossypium hirsutum* (*Moura et al., 2019*) and eight in foxtail millet (*Yadav et al., 2015*) and rice (*Kapoor et al., 2008*). In addition, five DCL proteins are encoded in maize and four in *Arabidopsis* (*Qian et al., 2011*). Furthermore, DCL transcription under abiotic stress treatments of salinity and drought has been reported (*Kapoor et al., 2008*). Genetic analysis of *Arabidopsis* revealed specialized and overlapping functions of DCL proteins (*Henderson et al., 2006*; *Fahlgren et al., 2007*); for example, AtDCL1 and AtDCL3 are involved in *Arabidopsis* flowering (*Schmitz et al., 2007*). In addition, it has been revealed that AtDCL2 and AtDCL4 share a functional overlap in an antiviral defense (*Deleris et al., 2006*), while AtDCL2, AtDCL3, and AtDCL4 exhibit overlapping functions in trans-acting siRNA (tasiRNA) and siRNA production and DNA methylation stability (*Henderson et al., 2006*).

The current research was conducted to determine the genome-wide identification and functional characterization of DCL protein families in okra plants relative to other members of the Malvaceae family by analyzing their evolutionary relationship, exon–intron structure, and conserved domains and motifs. In addition, the protein–protein interactions and expression of transcripts in different okra tissues under drought stress were analyzed to understand the DCL gene's functional role.

## MATERIALS AND METHODS

### Plant materials and stress treatment

The okra line OEL12 developed at the Applied Genomics and Molecular Biodiversity laboratory (Genetics Department, Faculty of Agriculture, Ain Shams University, Egypt) was used in this study. A total of 30 seeds were planted in a growth chamber. After 20 days of vegetative growth, the seedlings were subjected to dehydration. A water deficit was achieved by stopping irrigation entirely. The healthiest plants were selected in triplicate for the sampling of leaves and roots before dehydration as a control and then after 7 and 15 days of desiccation. Sampled tissues were collected in liquid nitrogen and directly used for RNA extraction.

### DNA extraction, library preparation, and genome sequencing

The total genomic DNA was extracted following the manufacturer's instructions from 2 g of fresh leaves from four okra plants using the WizPrep gDNA Mini Kit (Cell/Tissue) (Wizbio Solutions, Seongnam, Korea). The integrity, quality, and concentration of the DNA were determined by 1.0% agarose gel electrophoresis and a Quantus Fluorometer (Promega, Madison, WI, USA). We prepared approximately 300 bp of paired-end libraries using an Illumina TruSeq Library Preparation Kit (Illumina, San Diego, CA, USA). The libraries were sequenced in the paired-end mode, generating raw data of 150 bp on an Illumina Hiseq 4000 platform (Novogene, China), following the manufacturer's recommendations. Preprocessing was performed using Trimmomatic 0.39 software (*Bolger, Lohse & Usadel, 2014*), which removed adapter sequences, low-quality reads, and over-N-base reads. The quality of the clean short reads was assessed using FASTQC v0.11.9 (*Andrews, 2010*) and MULTIQC software (*Ewels et al., 2016*). Finally, the cleaned reads were assembled by mapping them to the reference genome (*G. hirsutum*, NAUNBI v1.1 assembly) using the Geneious assembler and mapper implemented in Geneious Prime (*Kearse et al., 2012*).

### Reference index

The DCL genome sequences of Malvaceae species were obtained from specialized online databases. The *Theobroma cacao* DCL gene copies (TcDCLs) were downloaded from MaGenDB (http://magen.whu.edu.cn), while the three cotton species *Gossypium arboreum* (BGI-CGB v2.0 assembly genome; GaDCLs), *Gossypium raimondii* (JGI assembly v2.0 data; GrDCLs), and *G. hirsutum* acc. TM-1 (NAUNBI v1.1 assembly genome; GhDCLs) were downloaded from the CottonGen (https://www.cottongen.org/) database. *Arabidopsis thaliana* Dicer-like protein sequences were obtained from TAIR database release 10 (http://www.arabidopsis.org).

## Okra DCL mapping and copy number assay

Since no reference genome was available for *A. esculentus*, DCL genes were assembled, identified, and annotated using a *de novo* assembly approach based on reads mapped to the DCL reference collection. The short reads were then remapped to the *de novo* assembled AeDCLs with 25-times iterations and only when paired-end reads were mapped. Geneious prime was used to perform the mapping and *de novo* assembly.

The newly assembled okra DCLs were tested for copy number because the DCL gene family is composed of the single-copy genes DCL1 and DCL4; in some organisms, including cotton, two or more copies can be found for DCL2 and DCL3 (*Liu, Feng & Zhu, 2009*; *Moura et al., 2019*). The total mapped reads were quantified and transformed by total read length. To estimate the copy number of the DCLs in the okra genome, the read depth from four DNAseq samples was applied. This quantified the expected number of gene copies per genome relative to the single-copy genes DCL1 and DCL4.

## Phylogenetic and evolution of malvaceae DCLs

The CDS was translated into the amino acid sequence (protein) following the standard translation table for each DCL. The AeDCLs and the retrieved AtDCLs, TcDCLs, GaDCLs, GrDCLs, and GhDCLs, protein sequences were aligned by MAFFT aligner (*Katoh & Standley, 2013*). The phylogenetic tree was built by FastTree V2 (*Kumar et al., 2018*) using the maximum likelihood method (ML) based on the Jones-Taylor-Thorton model with 1k bootstrap samples (default parameters).

## Okra DCLs protein analysis

In order to predict the DCLs protein structure, we used the protein homology/analogy recognition engine V 2.0 software (Phyre2; http://www.sbg.bio.ic.ac.uk/phyre2/). The conserved functional domains were identified through the InterProScan V2 plugin in Geneious (*Quevillon et al., 2005*). The secondary structure for each domain was tested and visualized using the SWISS-Model online tool (Expasy.org). To discover the motifs of DCLs, the MEME Suite web server V5.5.0 (*Bailey et al., 2015*) was used with the following parameters: the motif discovery mode was preset as "classic", the number of repetitions was set to any number, and motifs were limited to ten motifs. The physicochemical properties of the DCL proteins were characterized using the ProtParam online tool (Expasy.org), while the sublocalization prediction was performed using Cell-Ploc 2.0 (http://www.csbio.sjtu.edu.cn/bioinf/Cell-PLoc-2/). Promoter cis-regulatory elements were identified in 1500 bp upstream genomic regions of the translation start site using the PlantCARE bioinformatics tool (*Lescot, 2002*) and visualized by manual curation on Circos plot (*Krzywinski et al., 2009*). Protein-protein interactions of the DCL proteins and related functional enrichment analysis were mined in String-db (https://cn.string-db.org/) based on *G. raimondii* (Gr).

## RNA extraction and qPCR

Total RNA (including DNAse treatment) was extracted from plant tissues (root and leaf tissues) using the EasyPure® Plant RNA Kit (TransGen Biotech, Beijing, China), according to the manufacturer's instructions. The experiment involved three independent

biological replicates for each case. The quantity and quality of RNA extracts were assessed by electrophoresis on 2% agarose gels and the Quantus™ Fluorometer (Promega, Madison, WI, USA). EasyScript First-Strand cDNA Synthesis SuperMix Kit was used to synthesize cDNA from RNA extracts (TransGen Biotech, Beijing, China).

Based on the newly acquired DCL-CDS, specific primers for the qRT-PCR were designed by Primer3 software (*Untergasser et al., 2012*) implemented in Geneious Prime software. A list of the primer sequences can be found in Table S1. In order to verify the correct amplification of the desired fragments, all primer pairs were tested using conventional PCR and qRT-PCR using DNA extraction products and cDNA, respectively, with optimal annealing temperatures. TransStart® Green qPCR SuperMix was used for quantitative real-time RT-PCR (qPCR) (TransGen Biotech, Beijing, China). In triplicates, reactions were performed and contained 100 ng of cDNA, 0.5 µL of each primer (10 µM/µL), and 10 µL SYBR Green Master Mix in a final volume of 20 µL. The amplification PCR program was set as follows: 95 °C for 5 min, followed by 40 cycles of 95 °C for 15 s, 55 °C for 20 s, and 72 °C for 30 s. Melting curve analysis was performed by increasing the temperature from 55 to 95 °C (0.5 °C per 10 s).

The relative fold difference was calculated in each experiment using the $\Delta \Delta$ Ct method (*Livak & Schmittgen, 2001*). The GAPDH was selected as an internal control to normalize DCL Ct values in okra (*Cui et al., 2020*; *Ong et al., 2021*). Using Microsoft Excel, the DCL expression-based histograms were demonstrated, and the differences between cases were tested for significance using a one-tail paired $T$-test.

## RESULTS

### Okra DCL gene assembly and annotation

After filtering, the clean short reads mapped to the reference DCLs were reassembled *de novo* and annotated using the *ab initio* prediction method (*i.e.,* AUGUSTS) and BLAST. Complemented with a relatively equal number of mapped reads per reference, four single-copy high-quality DCL genes were successfully assembled. The lengths of the AeDCLs were 8,494, 5,214, 4,731, and 9,329 bp for AeDCLs 1, 2, 3, and 4, respectively. The number of detected exons varied from a single exon in *AeDCL3* to 24 exons in *AeDCL4*, while *AeDCL1* and *AeDCL2* recorded 20 and 18 exons, respectively (Fig. 1).

### Physiochemical characteristics of okra DCLs' predicted protein sequences

The predicted physicochemical properties of the newly assembled okra DCLs were analyzed. The results showed that the molecular weights of the okra DCL proteins were highest for AeDCL1 (225.341 kDa) and lowest for AeDCL2 (137.047 kDa), while the weights for *AeDCL3* and *AeDCL4* were 177.62 and 178.71 kDa, respectively. The approximate isoelectric point was 6. The points for *AeDCL1* and *AeDCL4* were 6.07 and 6.34, respectively, while *AeDCL2* had the highest point, at 6.55, and *AeDCL3* the lowest point, at 5.99. The hydrophobic amino acid frequency was less than half of the total amino acid length for all the AeDCL genes, and this indicated a high hydrophilicity property shown in negative values, confirming the hydrophilic nature of AeDCL proteins. With an instability index

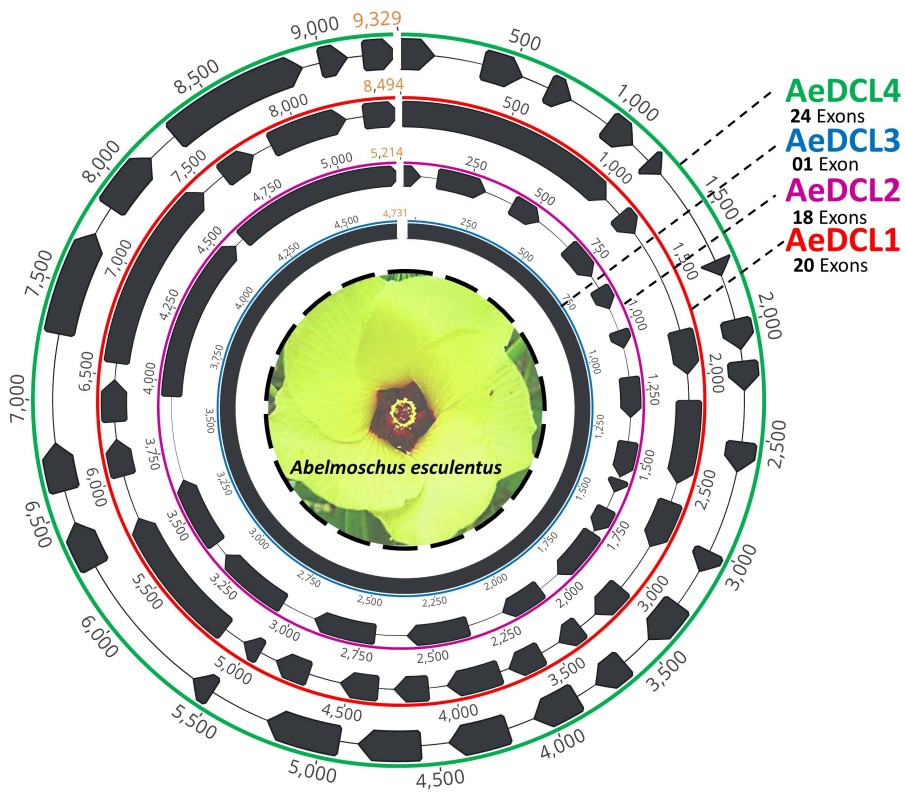

**Figure 1  Schematic drawing represents the four single-copy okra DCL genes *de novo* assembled in this study.** Each AeDCL showed different numbers of exons (marked in black arrows) interleaved with intronic regions, except for AeDCL3, where a single exon was detected with no introns.

**Table 1  Physicochemical characteristics for the amino acid translation of the okra DCLs.**

| Gene name | Length (aa) | Mw (kDa) | Isoelectric point | Hydro-phobic (aa) | Hydro-philicity | Instability index | Extinction coefficient |
|---|---|---|---|---|---|---|---|
| AeDCL1 | 2,008 | 225.346 | 6.07 | 941 | −0.385 | 40.26 | 202,485 |
| AeDCL2 | 1,207 | 136.979 | 6.55 | 558 | −0.150 | 41.19 | 141,585 |
| AeDCL3 | 1,576 | 177.276 | 5.99 | 722 | −0.143 | 42.05 | 127,905 |
| AeDCL4 | 1,589 | 178.658 | 6.34 | 735 | −0.168 | 50.44 | 129,310 |

of greater than 40, the okra DCL genes translated stable proteins, with factors ranging from 40.26 for *AeDCL1* up to 50.44 for *AeDCL4*. The model predicted that all the AeDCL proteins would be localized in a subcellular region of the cytoplasm, indicating that they had a non-secretory nature, and that all DCLs would most probably carry out metabolic activities within the cell (Table 1).

## Okra DCLs' secondary structures and conserved functional domains

The secondary structure of *AeDCL1* was predicted to be 40.98% alpha helix, 41.63% random coils, 13.16% extended strands, and 5.63% turns. For *AeDCL2*, the secondary structure was predicted to be 47.97% alpha helix, 33.31% random coils, 13.84% extended

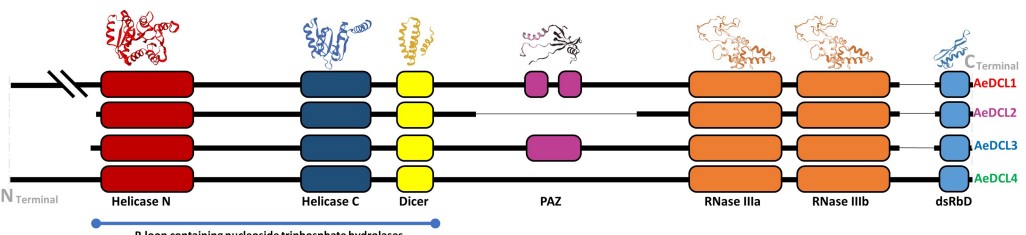

**Figure 2** Okra DCLs protein structure and conserved domains identified by SWISS-model and Inter-Pro database.

strands, and 4.89% turns; for *AeDCL3*, 43.85% alpha helix, 36.93% random coils, 14.47% extended strands, and 4.76% turns; and for *AeDCL4*, 45.81% alpha helix, 37.38% random coils, 12.84% extended strands, and 3.96% turns. In addition, AeDCLs had conservative functional domains of two DEAD-like helicase superfamilies, N (IPR006935) and C (IPR001650), one Dicer dimerization domain superfamily (IPR038248), one fragmented ribonuclease III domain (a-b; IPR000999), and one double-stranded RNA-binding domain (dsRBD; IPR014720). The PAZ domain (IPR003100) was recorded only for *AeDCL1* and *AeDCL3* (Fig. 2).

## Cis-element analysis of okra DCLs' promoter sequences

A total of 1,000 bp upstream sequences were successfully assembled for each AeDCL in order to analyze the cis-elements. As a result, 26 transcription factor–binding sites (TFBSs) were identified and enriched for all the AeDCLs. The TATA-box number of sites was found for all AeDCLs except for *AeDCL4*; the maximum number of TATA-box sites were found in *AeDCL2*, *AeDCL3*, and *AeDCL1*. In addition, the stress-related TFBSs were highlighted, the ABRE site regulated under ABA induction was found in the promoter regions of *AeDCL1* and *AeDCL4*, and the AUXRR-core site regulated by auxins was recorded only in *AeDCL3*. The MBS site regulated by drought stress was exclusively found in the *AeDCL1* promoter. However, the TC-rich site, regulated generally by stresses, was recorded in both the *AeDCL1* and *AeDCL3* promoters (Fig. 3).

## Evolution and phylogenetics of DCL gene family

Based on the DCLs' protein alignment in four Malvaceae species and the plant model *A. thaliana*, four highly supported clusters (*i.e.*, with bootstrap support values of 100) were observed, each represented by one DCL gene copy. The *A. thaliana* DCLs outgrouped each of the four clades, followed by *T. cacao*, the most distant Malvaceae species among the analyzed species. Okra DCLs followed *T. cacao* regarding gene copy number and distance compared with the three diverse cotton species *G. hirsutum, G. arboreum*, and *G. raimondii*, except for *AeDCL3*. One cotton species, *G. hirsutum*, is an allotetraploid that contains two diploid sets (A and D) besides the homologous copies (a and b) of DCL2 and DCL3. In one case, *AeDCL3* was clustered with the DCL3a subcluster with a high bootstrap value of 98. In the other case, *AeDCL2* was not clustered with either the a or b clades of the homologous

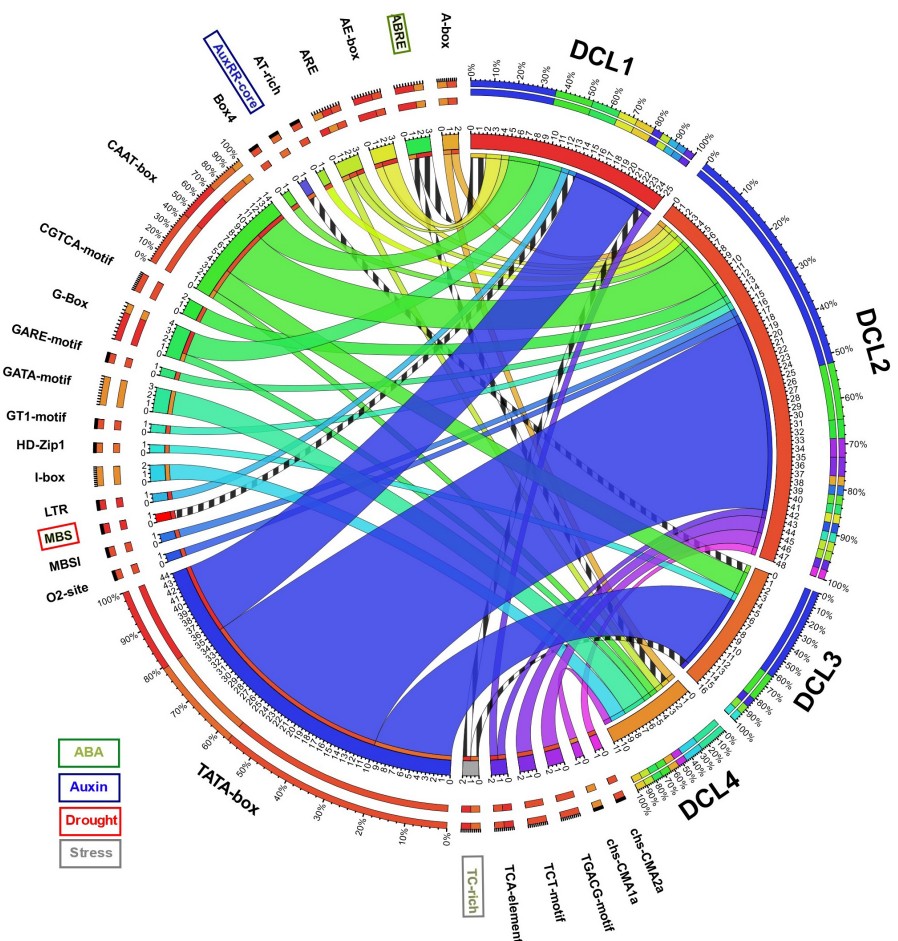

**Figure 3** **Circos plot showing number and types of transcription factor binding sites (TFBS) detected in the newly assembled promotor region (1,000 bp upstream) AeDCL genes.** Stress-related TFBS are highlighted while its presence and occurrence in AeDCLs are shown in stripped marks.

copies to DCL2. It is worth mentioning that DCL3 was not found in genome A of the *G. hirsutum* species (Fig. 4).

By comparing all Malvaceae DCLs included in the current study, seven conserved motifs were identified for all copies, except for GhDCL1 (genome D), GhDCL3 (genome D), GrDCL3, GaDCL2a, and GaDCL2b. All the AeDCLs had a conservative number of motifs, regardless of the DCL copy. Each of the conserved motifs was compared with the conserved domain position and sequence. In Fig. 4, from left to right, two can be seen as part of the helicase N domain, the third motif forms part of the helicase C domain, the fourth and fifth motifs are observed twice for each, and all are part of the two RNAse III domains. No motifs were detected for the Dicer, PAZ, or dsRNA-binding domains.

## Protein–protein interactions and gene ontology of Malvaceae DCLs

To study the interactions of DCLs with other gene families, protein–protein interaction data were mined for *G. raimondii* (Gr), the available representative genome in String-db,

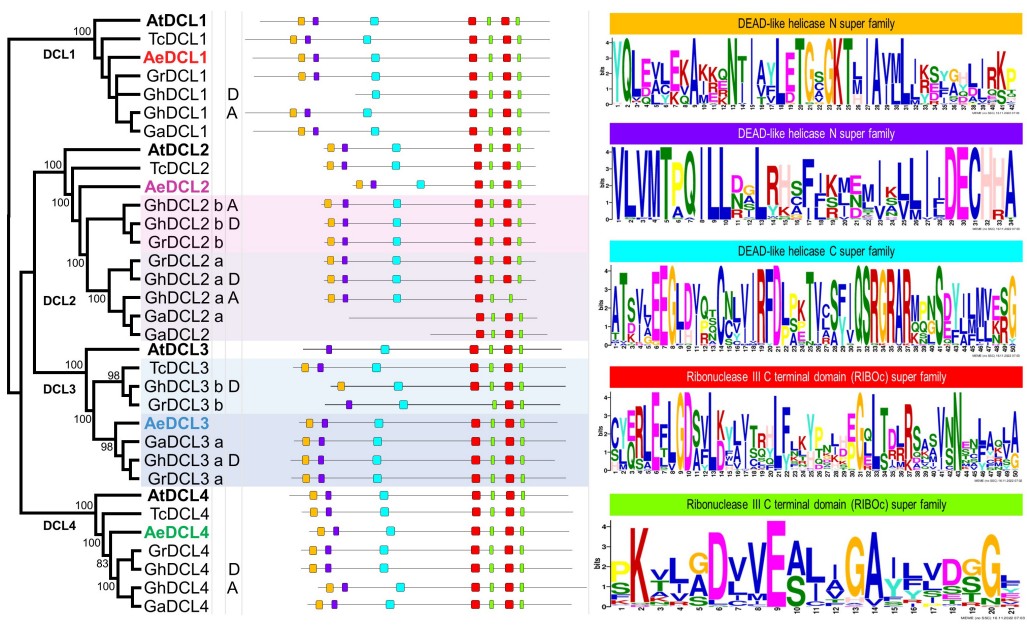

**Figure 4** **Phylogenetic tree and annotated motifs among the representative species of the Malvaceae family.** *Arabidopsis thaliana* (At), *Theobroma cacao* (Tc), *Gossypium hirsutum* (Gh), *G. arboreum* (Ga), *G. raimondii* (Gr) and *Abelmoschus esculentus* (Ae). Two motifs were observed as a part of the helicase N domain (orange and violet boxes), one motif formed part of the helicase C domain (turquoise box), the fourth and fifth motifs were observed twice for each, and all were part of the two RNAse III domains (red and green boxes).

followed by a functional enrichment analysis based on the interaction networks. Each of the DCL copies had a specific interactive network with other genes. DCL1 had an exchange interaction with the double-stranded RNA-binding protein 1-like (DRB1), Smad nuclear-interacting protein 1 (DDL), RNA-dependent RNA polymerase 2 (RDRP), small RNA 2′-o-methyltransferase-like (CRM2), Argonaute 2 (AGO2) and Argonaute 6 (AGO6) proteins, and serrate RNA effector molecule protein; all are involved in RNA and miRNA preprocessing and silencing and in RNA interference (Stringdb; CL:17564 and CL:17485). The DCL2 and DCL3 interactive networks were mainly related to Argonaute genes 1, 2, 5, 6, and 7. Both networks are involved in RNA-mediated post-transcriptional gene silencing (GO:0035194), RNA-mediated gene silencing (GO:0031047), and siRNA processing (GO:0030422). DCL4 had a hybrid interactive pattern in its relations to DCL1, DCL2, and DCL3; the DCL4 interacted with the AGO1, 2, 5, 6, and 7 proteins and the RDRP and DRB4 genes. All were involved in RNA and miRNA preprocessing and silencing and in RNA interference, as well as RNA-mediated post-transcriptional gene silencing (Fig. 5).

## Okra DCLs' expression analysis under drought stress

Quantitative polymerase chain reaction (qPCR) profiling of the four AeDCLs was performed under dehydration conditions to contrast the leaves and root tissues of the okra plants. The qPCR was normalized with the GAPDH housekeeping gene to the control
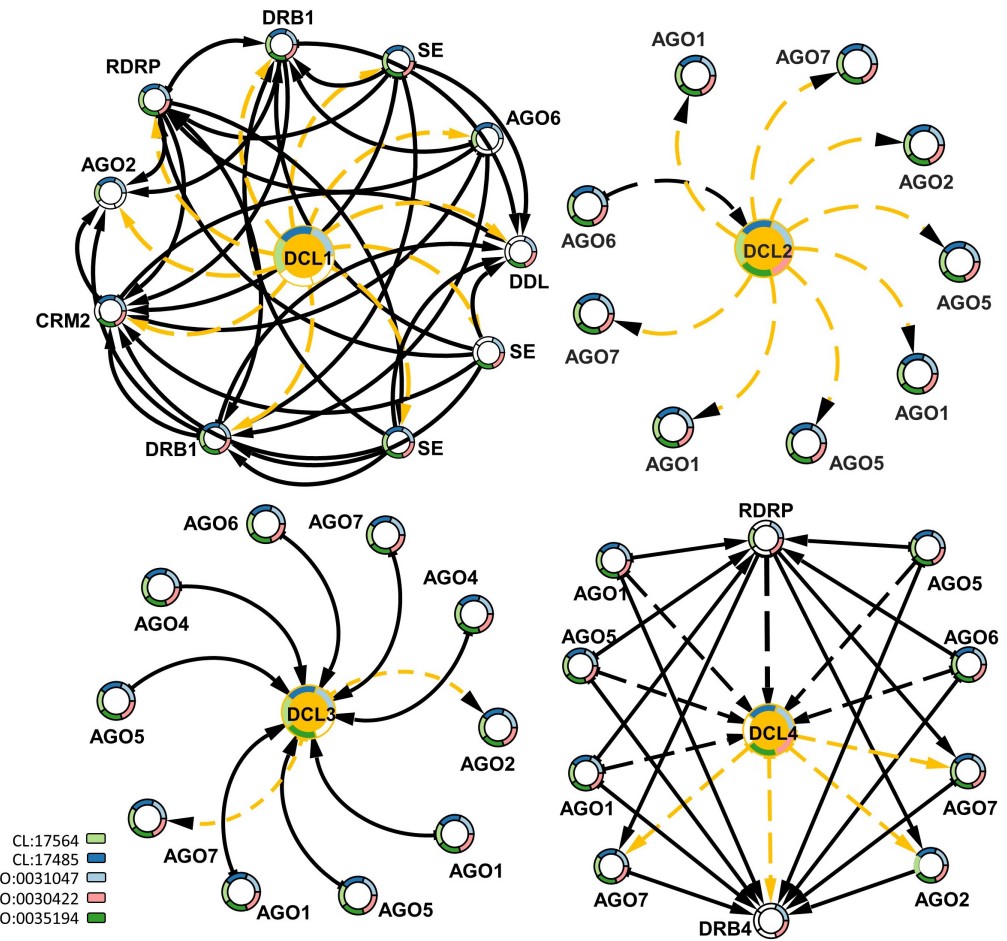

**Figure 5   Protein–protein interaction network enriched by GO ids for the four DCLs gene copies.** The dash-orange arrows represent the interaction direction from DCLs to other interacting genes, in contrast, the dash-black arrows represent the vice versa. Genes are circled and annotated by colors, each color representing an enriched pathway (CL) or gene ontology term (GO).

(hydrated) condition. All the AeDCLs were up-regulated under drought conditions, and a higher fold change was seen in their leaves than in their roots. *AeDCL4* leaves had the highest expressed DCL copy after 7 days of dehydration and the lowest after 15 days of dehydration. In terms of the roots, the *AeDCL1* roots had the highest expressed DCL copy after 7 days of dehydration and the *AeDCL3* had the lowest expressed DCL copy after 15 days of dehydration (Fig. 6).

An intriguing observation was made regarding the fold changes of *AeDCL1*, *AeDCL2*, *AeDCL3*, and *AeDCL4* in both the leaves and roots of the dehydrated plants. *AeDCL1* exhibited substantial increases in the fold changes of both its roots and leaves. After 7 days of dehydration, the fold change reached a value of 1 in both the leaves and roots, and after 15 days of dehydration, the fold change increased to a value of 2 in the leaves and a value of 8 in the roots.

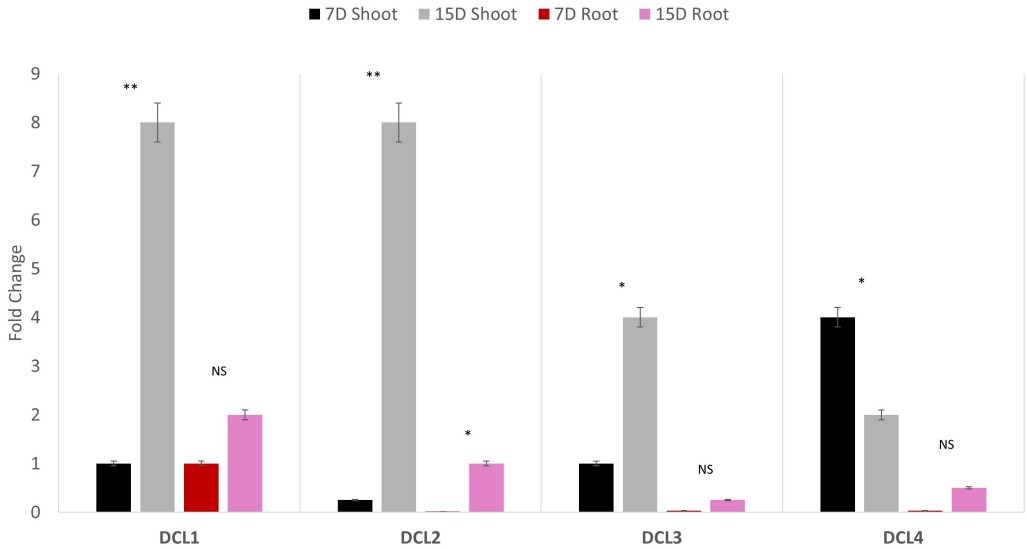

**Figure 6** **Quantitative PCR of four AeDCLs for okra plant leaves and roots sampled over 7 and 15 days (D) of dehydration.** Statistical differentiation between the quantified replicated for each gene per tissue was *<0.05, ** <0.01, and NS = not significant.

Similar patterns were observed for *AeDCL2* and *AeDCL3*, with higher fold changes in dehydrated leaves compared with dehydrated roots, although the rates of change were lower. After 15 days of dehydration, all AeDCLs displayed significant increases in the fold change, with leaves exhibiting higher fold changes than roots. Interestingly, *AeDCL4* demonstrated a distinct expression pattern. The expression was high in its leaves after 7 days of dehydration but significantly decreased in its leaves after 15 days of dehydration, contrasting with the expression in its roots (Fig. 6).

## DISCUSSION

In the present study, we assembled, characterized, and conducted a comprehensive survey to better understand the DCL protein family in Malvaceae, with a specific focus on the okra plant *A. esculentus*. Our investigation analyzed diverse data sources, including publicly available information, to gain a deeper understanding of the DCL proteins in this plant family. Our findings revealed an intriguing contrast between okra and other well-studied plants such as cotton, maize, poplar, and rice. Unlike these plants, okra did not possess any additional copies of DCL2 and DCL3 (*Moura et al., 2019*). This observation shed light on the evolutionary diversity and unique genomic characteristics of DCL genes in different plant species. Interestingly, the duplication of DCL3 has been reported as a distinct event in eudicots, whereas DCL2 duplication occurred independently in monocots and eudicots following their genetic separation (*Margis et al., 2006*). The extra DCL3, denoted herein as DCL3b, showed a complete DCL3 structure in *G. hirsutum* and *G. raimondii* but not in *G. arboreum*, in contrast to some previous reports (*Moura et al., 2019*). To further characterize the DCL genes in okra, we examined the number of exons and introns across different

species within the Malvaceae family. Our results demonstrated a significant variation, with the number of introns ranging from 7 to 25 in different cotton species (*Moura et al., 2019*) and a wide range of numbers of DCLs in okra, from a single intronless copy (*AeDCL3*) to 23 introns in *AeDCL4*. Despite these variations, the open reading frames and predicted secondary structures of the DCL copies remained relatively conserved, indicating functional similarity across the analyzed sequences.

The secondary structure was correctly predicted and showed the typical DCL protein structures. Dicer and DCL proteins are large multi-domain ribonucleases. Vertebrate, insect, and plant Dicer and DCL proteins generally contain six domains: the DEAD-box, helicase-C, Dicer (DUF283), PAZ, RNase III, and dsRBD domains (*Margis et al., 2006*). In eukaryotes, the DUF283 domain might play a role in siRNA/miRNA strand selection by directly sensing asymmetry in RNA duplexes or recruiting another dsRBD protein (*Dlakić, 2006*). The PAZ, RNase III, and dsRBD domains are responsible for dsRNA binding and cleavage. The AeDCL gene copies were completely assembled and annotated as the reference. Even though the PAZ domain was annotated in all the cotton DCL copies (*Moura et al., 2019*), the PAZ domain was recorded only for *AeDCL1* and *AeDCL3*. Dicer's PAZ domain binds to the RNase IIIa domain through a long α helix and is capable of binding dsRNA endings with a 3′ two-base overhang (*MacRae et al., 2006*) and may also play a role in binding single-stranded RNAs (*Kini & Walton, 2007*). A further analysis is required to fully understand the changes in *AeDCL2* and *AeDCL4* protein translation in the PAZ domain site and the nature of those differences in terms of their functional activity in okra.

Because of the importance of miRNA/siRNA biogenesis, the Dicer and DCL proteins are essential for eukaryote development (*Yang et al., 2005*) and apoptosis (*Matskevich & Moelling, 2008*). In *A. thaliana*, the four DCL proteins play different roles (*Xie et al., 2004*; *Moissiard et al., 2007*; *Mlotshwa et al., 2008*). Besides its role in miRNA production, DCL1 is also involved in producing small RNAs from endogenous inverted repeats. The other three DCLs are siRNA-generating enzymes (*Liu et al., 2007*). It is believed that DCL1 and DCL3 work together to promote flowering in *Arabidopsis* plants (*Schmitz et al., 2007*). As we analyzed the DCL transcripts in leaves and roots suffering from dehydration, we observed that all AeDCLs appeared to be functional and modulated. That was contrary to what was previously reported for other plant species (*Belal et al., 2022*). One different aspect is the lack of homologous copies in okra, a situation that is similarly found in grapevines (*Zhao et al., 2015*), chickpeas, pigeon peas, *Arachis ipaensis* (*Garg et al., 2017*), and peppers (*Qin et al., 2018*).

The other different aspect of okra is that the fold changes of AeDCL leaves were higher than the changes in their roots. For example, DCL1 was highly up-regulated in the roots of the commercial cotton cultivar Delta Opal, and DCL1 and DCL3b in the roots of the Fibermax 966 (FM) cultivar (*Moura et al., 2019*). In the peach plant *Prunus persica*, the highest expression pattern was found for DCL2 copies (a and b) in root tissues, while DCL1 had the lowest expression pattern of all the other tissues (*Belal et al., 2022*). The *AeDCL1* and *AeDCL2* were highly up-regulated in their leaves and roots and significantly increased over time, suggesting an overlapping functional pattern in leaves and different regulatory mechanisms under dehydration conditions than other plants. Regarding the antiviral

defense, DCL2 and DCL4 shared an overlap function (*Deleris et al., 2006*), and DCL2, DCL3, and DCL4 engaged in siRNA and tasiRNA production, as well as DNA methylation establishment and maintenance (*Henderson et al., 2006*). The presence or absence of extra copies of DCL2s and DCL3s in cotton in a heterologous system will need to be further investigated to understand the effects of these DCLs on RNA silencing machinery in okra.

The current study annotated two drought-related cis-elements: the MBS-site, which is exclusively regulated by drought stress in the *AeDCL1* promoter, and the ABA-responsive cis-acting element ABRE site, which was found in the DCL1 and DCL4 promoters. Dicer expression studies in *Arabidopsis* have shown that the loss of function of all four DCLs causes ABA hypersensitivity during seed germination. This may be due to the biogenesis of one or more particular microRNAs that play a role in ABA signaling (*Zhang et al., 2008*). The DCL genes were part of a functional interactive network, including ARGONAUTE (AGO) and RNA-dependent RNA polymerase (RDR). According to the literature, the long non-translated RNA (lsiRNA) and the viral-small RNA (vsiRNA) generated by DICER cutting are subject to the incorporation of small RNAs into AGO-containing RNA-induced silencing complexes (RISCs) to confer sequence specificity in the silencing of RNA information (*Rivas et al., 2005*; *Aliyari & Ding, 2009*; *Borges & Martienssen, 2015*). In addition, plants and fungi use RNA-dependent RNA polymerase (RDR) to convert aberrant RNA into dsRNA, amplifying small RNAs and intensifying RNA silencing (*Wang et al., 2010*; *Borges & Martienssen, 2015*). The functional network involves RNA and miRNA preprocessing, RNA-mediated post-transcriptional gene silencing, RNA interference, and the enriching of the viral silencing and resistance mechanisms in plants. In Egypt, okra plants suffer from a lethal seed-borne virus known as okra leave curl virus (OLCV) (*Magdy et al., 2021*). A viral-induced expression analysis of okra plants using RNAseq and qPCR analysis is planned to investigate the role of DCL, AGO, and RDR genes in regulating the OLCV defense mechanism of the plants. This strategy will deeply examine DCL modulation by silencing each DCL copy and/or it will be used in combination with the VIGS technique to analyze OLCV-tolerant okra lines produced in our laboratory.

## CONCLUSION

This study was a comprehensive investigation of the DCL protein family in the okra plant *A. esculentus* in the context of the Malvaceae plant family. Several key findings have emerged through genome-wide identification, phylogenetic analysis, structural characterization, cis-element analysis, protein–protein interaction analysis, and gene expression profiling under drought stress. Okra's unique DCL gene architecture differs from well-studied plant species such as cotton, maize, and rice. Unlike these species, okra does not have additional copies of DCL2 and DCL3, highlighting the evolutionary divergence of DCL genes among plant lineages. Despite variations in introns number among different DCL genes, the open reading frames and secondary structures of okra DCLs remain relatively conserved. Notably, the presence of conserved functional domains, such as DEAD-box, helicase-C, RNase III, and dsRBD, reinforces the essential roles of these proteins in RNA processing and silencing. Analysis of promoter sequences revealed the presence of stress-related

cis-elements, such as the ABRE (ABA-responsive element) and MBS (MYB-binding site). These elements suggest potential involvement in drought stress responses, which aligns with the relative drought tolerance observed in okra. Phylogenetic analysis indicated the evolutionary relationships of DCL genes within Malvaceae species and provided insights into the gene duplication events in cotton genomes. As compared with cotton, the absence of certain homologous copies in okra highlighted the diversity in DCL gene copy numbers among plant species. The interactive networks of DCL proteins with other genes, particularly AGO (Argonaute) and RDR (RNA-dependent RNA polymerase), suggest their critical roles in RNA-mediated gene silencing pathways. These interactions underscore the importance of DCL proteins in regulating gene expression and defense mechanisms against viral infections. Expression analysis of DCL genes in okra under dehydration conditions revealed significant up-regulation in response to drought stress. Interestingly, the fold change in the okra leaves was higher than in their roots, indicating overlapping functional patterns in leaves and dynamic regulatory mechanisms in response to dehydration. The findings from this study contribute to the broader knowledge of plant stress responses and have implications for crop improvement strategies, particularly in developing more resilient okra varieties in the face of changing climatic conditions.

### Funding
This work was supported by Princess Nourah bint Abdulrahman University Researchers Supporting Project number (PNURSP2024R402), Princess Nourah bint Abdulrahman University, Riyadh, Saudi Arabia. The funders had no role in study design, data collection and analysis, decision to publish, or preparation of the manuscript.

### Grant Disclosures
The following grant information was disclosed by the authors:
Princess Nourah bint Abdulrahman University Researchers Supporting Project: PNURSP2024R402.

### Competing Interests
Diaa Abd El-Moneim is an Academic Editor for PeerJ.

### Author Contributions
- Hagar Tarek Elhefnawi performed the experiments, analyzed the data, authored or reviewed drafts of the article, and approved the final draft.
- Mohamed Abdel Salam Rashed conceived and designed the experiments, authored or reviewed drafts of the article, and approved the final draft.
- Ayman Atta conceived and designed the experiments, authored or reviewed drafts of the article, and approved the final draft.
- Rana M. Alshegaihi performed the experiments, authored or reviewed drafts of the article, and approved the final draft.

- Khairiah Mubarak Alwutayd performed the experiments, authored or reviewed drafts of the article, and approved the final draft.
- Diaa Abd El-Moneim analyzed the data, authored or reviewed drafts of the article, and approved the final draft.
- Mahmoud Magdy conceived and designed the experiments, performed the experiments, analyzed the data, prepared figures and/or tables, authored or reviewed drafts of the article, and approved the final draft.

## Data Availability

The four AeDCLs gene copies are available at NCBI: OQ127274, OQ127275, OQ127276 OQ127277.

## Supplemental Information

Supplemental information for this article can be found online at http://dx.doi.org/10.7717/peerj.16232#supplemental-information.

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
