# Peer review of "Genomic assembly, characterization, and quantification of DICER-like gene family in Okra plants under dehydration conditions"

_PeerJ, doi:10.7717/peerj.16232_

## Round 0.1 · original submission · Minor Revisions

Dear Authors,
Your manuscript is very valuable in terms of elucidating the genetic mechanism of drought stress in a less studied plant. Although it is well written in general terms, it requires a few corrections. I believe your correction, taking into account the criticism of the reviewers, will improve it even more. I suggest it be revised with more current literature in the Discussion section.
Best regards

Reviewer 1 ·

Basic reporting

This paper is generally well organized and written. Its relevance to the world's bleeding wound, drought, is appreciated.
There are several types of drought. biotic drought (caused by Fusarium and verticillium dahliae Kleb.) and abiotic drought. the first paragraph of the introduction mentions which drought.

What concrete role could the dicer-like (DCL) gene play in combating this drought? The possibility of transferring this gene through hybridization?
line 47-48. 8 years have passed since then. Can you give an updated reference on the impact of drought on okra today? Climate events that change daily or even hourly require us to provide the most up-to-date data.

Experimental design

Please indicate the ancestor of 12 okra lines. What kind of agricultural properties of ancestor used to obtaine the lines. and please provide the lines segregation levels. Like f2,or f3 or later?

Line 91-92 and 93. Why 20 days? is there any studies that 20 days are enough? Did you measured?
Line 95-96. Did you picked up young leaf only for RNA, What about DNA?

Have you done the integrity and optium concentration tests of the RNA? If so, how will it be? One of the robustness tests of RNA is to run it on an agarose gel.

According to which technique you did the extraction of DNA, for example, CTAP is one technique.
Remains Material & methods are amasing except there is no any statistical analysis chapter in this section

Validity of the findings

Line 182-185 this sentences until point is like metods passage. We prefer only results or combined results and discussion in this section?
Line 2002-2012 passage. There is neither citation nor any references from this study.

The rest of the results section was well constructed. I believe that it will increase its effectiveness if it is supported with literature in places. The summary of the literature in the Discussion section should be changed. it looks like an introduction. The similarities or differences between the results found by the researchers in this study and those found in the literature should be given by comparing them with each other. The citation technique in the discussion section needs to change. It is useful to look at the discussion section of any article in NCBI.

Additional comments

We fully support molecular studies, but the fight against drought requires studies that will take concrete steps in terms of the urgency of concrete applications. however, this article contains an admirable study. I think that such studies should be carried out on new and undiscovered plants, not on well-studied plants, and that presenting a feature that people can use will provide more service in today's world where food crisis and food security are problems.

·

Basic reporting

Dear authors, I would like to thank you for your manuscript submission, also for presenting an interesting work, especially for researchers working in Okra. However, some minor edits, and extensive English checking is required, For example,
Line 107 It seems that the verb (were) does not agree with the subject.
Line 133, the noun phrase amino acid sequence seems to be missing a determiner before it.
Line 134, the sentence contains a series of three or more words, phrases, or clauses, please consider inserting a comma to separate the elements.
Line 146, The word set appears repeatedly in this text. Consider using a synonym in its place.
Line 164, conjunction use may be incorrect in the sentences.
Line 165, and 168 contain an article usage problem.
Line 173, consider adding the missing verb in the sentences.

Experimental design

Experimental design, methods, and results are presented and listed in good homogeneity. Otherwise, in the material and methods part, more details about the plant developed line are required, also the detailed procedure of growth condition and stress treatment.

Validity of the findings

The subject of studying the gene family is vital, and yet relatively few studies have been conducted in Okra plants. The validity of a research study refers to how well the results among the study participants represent findings among similar individuals outside the study.

Additional comments

I recommend a minor revision, and I believe that the authors will consider all suggestions and that the manuscript will contribute to the field of Okra plant research.

Reviewer 3 ·

Basic reporting

I read the present manuscript with very interest which reports on the Genomic assembly, characterization, and quantification of DICER-like gene family in Okra plants under dehydration conditions. The authors identified four DCL single-copied genes in okra and characterized them at the molecular level. Subsequently, the authors also assessed the expression of all four AeDCLs under dehydration conditions in the leaves and root tissues of the okra line OEL12 developed by Ain Shams University, Egypt. In the future, authors also intend to investigate the role of DCL, AGO, and RDR genes in regulating the Okra leaf curl virus (OLCV) defense mechanism because OLCV is a major threat to Okra cultivation in Egypt. This study is very informative for researchers, particularly for Okra scientific community.

Major Comments:
1. In line 22: I’ll suggest to replace the horticulture word with vegetable.
2. In line 25: Perhaps authors want to suggest Okra crop as model not Okra plants….
3. In line 47: Please insert word stage after pod formation
4. In Line 79: Please write the full term of tasiRNA for the first time.
5. In Line 113: Please include the complete scientific name of T. cacao when it is first mentioned in the text.
6. Line 190-201: Please try to avoid abrupt statements, because the physiological properties of DCLs proteins are predicted using bioinformatics tools. Use the same format like digits after decimal for the molecular weight of each DCL protein. Similarly, the localization of AcDCLs proteins at subcellular levels is not demonstrated, it is only predicted based on Cell-Ploc 2.0 bioinformatics tools. Therefore, please avoid direct statements and rephrase the sentence in Lines 200-201
7. In Line 202: Please reference Fig. 2 in the text.
8. In Line 212: Please add the double-strand RNA-binding domain abbreviation as you mention in Fig. 2.
9. In Figure 4, I recommend that you briefly describe the motifs in the legend for the figure. This will enable readers to understand the figure without referring to the text.
10. In line 269-276: Please clarify and rephrase the sentences in a better way. For instance: the root reached one fold after seven days of dehydration in both leaves and roots…………..
11. In Line 347: OLCV means Okra leaf curl virus or Okra curl leaf virus ??

Experimental design

No comment

Validity of the findings

No comment

Additional comments

Please improve the language of the manuscript as mentioned earlier in basic reporting

---

## Round 0.2 · Minor Revisions

Dear authors,
I see you have revised your manuscript following the reviewers' recommendations. For the parts where you did not make any changes, you gave your answers by providing valid reasons.

Before acceptance, the writing needs to be improved, ideally by professional proofreading. For example, some phrases are written in telegram style (without a verb): 'Famous for its ability to tolerate long desiccation periods.'

Also, the logical order needs to be improved and you should add a Conclusion section

Best regards

**Language Note:** The Academic Editor has identified that the English language must be improved. PeerJ can provide language editing services - please contact us at copyediting@peerj.com for pricing (be sure to provide your manuscript number and title). Alternatively, you should make your own arrangements to improve the language quality and provide details in your response letter. – PeerJ Staff

---

## Round 0.3 · accepted · Accept

I think the language of the manuscript is now sufficient for publication.